# Comparing Targeting Strategies for Maximizing Social Welfare with Limited Resources

**Vibhhu Sharma & Bryan Wilder**
Machine Learning Department
Carnegie Mellon University
Pittsburgh, PA 15232, USA
{vibhhus,bwilder}@cs.cmu.edu

## Abstract

Machine learning is increasingly used to select which individuals receive limited-resource interventions in domains such as human services, education, development, and more. However, it is often not apparent what the right quantity is for models to predict. Policymakers rarely have access to data from a randomized controlled trial (RCT) that would enable accurate estimates of which individuals would benefit more from the intervention, while observational data creates a substantial risk of bias in treatment effect estimates. Practitioners instead commonly use a technique termed "risk-based targeting" where the model is just used to predict each individual's status quo outcome (an easier, non-causal task). Those with higher predicted risk are offered treatment. There is currently almost no empirical evidence to inform which choices lead to the most effective machine learning-informed targeting strategies in social domains. In this work, we use data from 5 real-world RCTs in a variety of domains to empirically assess such choices. We find that when treatment effects can be estimated with high accuracy (which we simulate by allowing the model to partially observe outcomes in advance), treatment effect based targeting substantially outperforms risk-based targeting, even when treatment effect estimates are biased. Moreover, these results hold even when the policymaker has strong normative preferences for assisting higher-risk individuals. However, the features and data actually available in most RCTs we examine do not suffice for accurate estimates of heterogeneous treatment effects. Our results suggest treatment effect targeting has significant potential benefits, but realizing these benefits requires improvements to data collection and model training beyond what is currently common in practice.

## 1 Introduction

Policymakers often face the difficulty of allocating a resource-limited intervention with the goal of targeting the intervention towards those who will benefit most from it. Indeed, the causal inference literature documents that any given treatment may not have the same effect on every individual that receives it (Wager & Athey, 2018; Künzel et al., 2019; Varadhan & Seeger, 2013). When there are observable features that correlated with greater benefit from the treatment, such variation can be used for targeting. Heterogeneity of treatment effect (HTE) refers to this nonrandom, explainable variability in the direction and magnitude of treatment effects for individuals within a population. Given this variability, policymakers often face the problem of selecting who to treat when having to assign a particular treatment to a group of people under a fixed budget. Machine learning methods seem to offer the promise of discovering richer forms of heterogeneity, allowing more effective targeting of interventions in practice.

The main challenge is that heterogeneous treatment effects are difficult to learn: doing so requires a potentially large amount of data that is *unconfounded*, i.e., where treatment is assigned in a manner (conditionally) independent of each individual's outcomes. Conducting a randomized controlled trial (RCT) to achieve this is often infeasible due to time constraints, resource limitations, and ethical

concerns. Policymakers may prefer to prioritize access to treatment via an already-available proxy metric that is believed to align with need if experimenting to gather additional evidence would be seen as unethical.

One particularly common proxy is to target according to the *baseline* risk each individual faces, i.e., their expected outcome in the absence of treatment (as opposed to the treatment effect, which is the difference in outcomes induced by treatment). This strategy has been referred to as *risk-based targeting* (Wilder & Welle, 2024). Individuals with poor predicted baseline outcomes may be seen as needing assistance the most. Policymakers may also believe that these individuals also stand the most to benefit since their status-quo prognosis is the worst. Importantly, baseline risks can often be learned using existing administrative data (from before a treatment was introduced) instead of requiring a new experiment, making this strategy easily implementable in many practical settings. For all of these reasons, risk-based targeting has seen widespread use by policymakers in a wide range of domains, including targeting humanitarian assistance (Aiken et al., 2021), allocating homelessness services via vulnerability scores (Shinn & Richard, 2022), and the use of "early warning systems" in education (Perdomo et al., 2023).

Despite this widespread usage, there is only a limited amount of work which empirically assesses the effectiveness of risk-based targeting: do individuals with the greatest baseline risk actually tend to benefit the most from intervention? Existing studies speak only to a single specific application domain each, for instance, targeting students for sending reminders for financial aid applications (Athey et al., 2024), preventing customer churn (Ascarza, 2018), or targeting cash transfers (Haushofer et al., 2022). (Kube et al., 2022) examines this from a human perspective by empirically eliciting decision-maker preferences for whether to prioritize more vulnerable households or households who would best take advantage of more intensive interventions when determining which homeless households to serve with limited housing assistance. (Ascarza, 2018) encourages practitioners to use RCTs to better inform their decision, however, as discussed above, running a RCT may be infeasible in many settings. The alternative more likely to be available to practitioners is to simply estimate treatment effects using observational data which likely suffers from confounding, potentially leading to biased estimates of treatment effects.

How should practitioners navigate this tradeoff between a more easily-learnable label that may not always correlate with benefit from an intervention (baseline risk) and a difficult-to-learn quantity (heterogeneous treatment effects) that captures the impact of the intervention? This corresponds to the choice of the appropriate target for prediction, as opposed to the specific model used to make the prediction. The choice of outcome variable has been observed to exert a disproportionate influence on the impacts of machine learning systems in many settings (Obermeyer et al., 2019; Coster, 2013; Gerdon et al., 2022). In the setting of targeting interventions for causal impact, practitioners have little empirically-grounded guidance. Our goal in this work is to inform the selection of an objective function for machine-learning based targeting of scarce interventions. We make three contributions towards this goal:

*First*, we assess the efficacy of risk-based targeting on a wider variety of real RCT datasets encompassing settings in economics, healthcare and education in contrast to prior studies that generally focus on one dataset. We find a generally noisy and variable relationship between baseline risk and treatment effects: high-risk individuals seem to benefit more on average in most domains, but with substantial variance in treatment effects which is not explained by baseline risk. When treatment effects can be estimated reliably (which we simulate by allowing the model to partially observe outcomes in advance), targeting based on these estimates produces substantially better results compared to risk-based targeting. However, in practice, treatment effects must be predicted from features alone, and limited data can make it difficult to learn accurate mappings from features to treatment effects, potentially degrading the performance of treatment effect based targeting.

*Second*, we compare risk-based targeting to targeting policies based on biased estimates of treatment effect obtained from confounded data. Such biased estimates are likely when conducting a full-fledged RCT is infeasible and policymakers have to rely on available observational data alone. Accordingly, potentially-biased causal estimates represent the likely alternative to risk-based targeting in many domains of practical interest. To our knowledge, these two strategies have not been explicitly compared. We find that across even relatively severe levels of confounding, a utilitarian policymaker often prefers targeting according to biased estimates of the treatment effect rather than baseline risk when treatment effects can be estimated reliably at no confounding.

*Finally*, we analyze the setting where a policymaker has inequality-averse preferences: oftentimes, policymakers may prefer interventions which benefit those who are worse-off to begin with even if they produce less aggregate impact. Such normative goals are one possible justification for risk-based targeting, even if risk-based targeting is less attractive in standard utilitarian terms. We compare the two targeting strategies under popular classes of social welfare functions which capture inequality-averse preferences. We find that when treatment effects can be estimated reliably, treatment effect based targeting is typically favorable to risk-based targeting, even in scenarios where policy makers are inequality-averse and the only available data is confounded to some degree.

All together, our results suggest that the barrier to targeting on treatment effects is more likely to lie in the collection of large and rich enough datasets to support accurate prediction of outcomes, while confounding or inequality-averse preferences have more limited impact. This implies that large-scale administrative datasets may be a more attractive avenue for policy learning than previously thought, even if they lack the guaranteed validity of a randomized controlled trial.

**Related Work:** Measuring heterogeneity in treatment effect and choosing which subpopulations to assign a treatment to has long been an active avenue of research in causal inference literature with a variety of methods proposed to solve this problem. (Green & Kern, 2012; Hill, 2011; Hill & Su, 2013; Foster et al., 2011; Wager & Athey, 2018) use forest-based algorithms to identify groups that show heterogeneity in treatment effect with other identified groups. (Tian et al., 2014) proposed to measure the interaction between treatment and covariates by numerically binarizing the treatment and including the products of this variable with each covariate in a regression model. (Künzel et al., 2019) uses meta-learners that decompose estimating the CATE into several sub-regression problems that can be solved with any regression or supervised learning method. The problem of choosing who to treat is closely related to identifying the heterogeneity in treatment effects. This often involves balancing policies based solely on estimates of conditional average treatment effect (CATE) with additional prioritization rules set by the policymaker. (Yadlowsky et al., 2021) proposes rank-weighted average treatment effect metrics for testing the quality of treatment prioritization rules, providing an example involving optimal targeting of aspirin to stroke patients. Meanwhile, (Gupta et al., 2020) provides a theoretical analysis of risk-based targeting strategies, establishing performance bounds for scoring rules when treatments are benign.

## 2 PRELIMINARIES

Consider a setting where there is a population of individuals who are candidates for a treatment or intervention. Each individual has a feature vector $X \in \mathbb{R}^d$. Here we are concerned with binary treatments. Following Neyman and Ruben's (Splawa-Neyman et al., 1990; Rubin, 1974) potential outcomes framework, we use $Y^{(1)}$ to denote the outcome that an individual would experience under treatment and $Y^{(0)}$ to denote the outcome they would experience if not treated. Their individual treatment effect is $Y^{(1)} - Y^{(0)}$. We assume that $(X, Y^{(0)}, Y^{(1)})$ are drawn i.i.d. for each individual from some joint distribution. In order to identify individuals who are likely to benefit, a common strategy is to use individuals' observed covariates to predict the expected treatment effect. The conditional average treatment effect (CATE) is defined as:

$$\tau(x) = \mathbb{E}\left[Y^{(1)} - Y^{(0)} \middle| X = x\right]. \tag{1}$$

Estimating the CATE in order to target based on treatment effects is a difficult statistical problem. Suppose we have access to data corresponding to $n$ people, labeled $i = 1, ..., n$, consisting of features $X_i$, a treatment assignment $W_i \in \{0, 1\}$, and the observed outcome $Y_i = Y_i^{(W_i)}$. Importantly, for each individual, we can observe only the outcome corresponding to the treatment they were actually assigned. Accordingly, identifying treatment effects typically requires a a no-unobserved-confounders assumption (Rosenbaum & Rubin, 1983): $\{Y^{(0)}, Y^{(1)}\} \perp\!\!\!\perp W \mid X$. This assumption is most credible in the context of a randomized controlled trial (RCT). In an RCT, the assignment of treatment, represented by $W_i$, is assigned independently of the potential outcomes $Y_i$ (potentially after stratification on covariates $X_i$). When data is purely observational, practitioners typically try to select a sufficiently rich set of covariates $X$ such that all potential confounders between outcomes and treatment assignment are measured. However, ensuring that all confounders are completely captured is notoriously difficult in practice, creating the likelihood that some bias in the estimated CATE remains (LaLonde, 1986; Pearl, 2009; Skelly et al., 2012; Milli et al., 2022).

As an alternative to targeting on treatment effects, policymakers often decide to treat people who are more vulnerable or worse-off at present, without attempting to quantify the benefit these individuals receive from treatment. This is quantified via a 'baseline risk'; we refer to the resulting allocation strategy as 'risk-based targeting'(Wilder & Welle, 2024). Baseline risk may sometimes be directly measured quantity (one of the covariates in $X$, for example baseline test scores in an educational context). In many settings though, it is estimated using a predictive model that uses the covariates as input. Let $b$ be a function that maps a set of covariates to a baseline risk measurement such that $b(X)$ represents the baseline risk and $b(X_i)$ denotes the baseline risk associated with $i$. Then this method involves selecting individuals with the highest values of $b$ for treatment, implying that these individuals have the highest 'risk' associated with them, which needs immediate resolution. It is important to note that this strategy requires only data on baseline outcomes prior to program implementation, with no information about the treatment's effect being incorporated in the policymaker's decision.

## 3 METHODS

We compare risk-based and treatment effect-based targeting strategies using real-world RCTs, which allow credible estimation of heterogeneous treatment effects because of the assumption of no unmeasured confounding. We simulate these targeting strategies under varying degrees of confounding and under different social welfare functions for policymakers, evaluating their effectiveness against randomization-enabled treatment effect estimates. We now detail the methodology used in each step of this process, starting with the datasets used.

### 3.1 DATASETS

We conduct experiments on a variety of RCTs across different domains as detailed below:

**Targeting the Ultra Poor (TUP) in India** ((Banerjee et al., 2021)): This RCT was conducted to study the long-term effects of providing large one-time capital grants to low income-families and observing how family income and overall consumption changed over a period of 7 years. We consider a family's total expenditure as the outcome, which is positively affected by treatment.

**NSW (National Supported Work demonstration) Dataset** ((Dehejia & Wahba, 1999; 2002; LaLonde, 1986): This study estimated the impact of the National Supported Work Demonstration, a job training program, on beneficiaries' income in 1978. We consider an individual's income in 1978 as the outcome, which is positively affected by treatment.

**Postoperative Pain Dataset:** Patients undergoing operations like tracheal intubations often experience throat pain following treatment (Mchardy & Chung, 1999). This RCT was conducted to test the efficacy of a licorice solution at reducing postoperative sore throat. The outcome we focus on is a patient's throat pain 4 hours after surgery. Here, the effect of the treatment is to reduce the amount of throat pain, hence the treatment effect is negative. In order to maintain consistency with other plots, we present results with the sign for treatment effect reversed.

**Acupuncture Dataset:** (Vickers et al., 2004) This RCT aimed to determine the effect of acupuncture therapy on headache severity in patients with chronic headaches. Our outcome variable is headache severity 1 year post-randomization. Here again, the effect of the treatment is to reduce the severity of headaches, hence the treatment effect is negative. In order to maintain consistency with other plots, we present results with the sign for treatment effect reversed.

**Tennessee's Student Teacher Achievement Ratio (STAR) project** (Achilles et al., 2008): This four-year study by the Tennessee State Department of Education examined class size effects on student performance. The design compared: 1) Small classes (13-17 students per teacher), 2) Regular classes (22-25 students), and 3) Regular classes with a teacher's aide. We focus only on the first two configurations to maintain binary treatment consistency with other RCTs. Our outcome measure is kindergarten students' cumulative test scores.

For each of these datasets, we estimate $E[Y^{(0)}|X]$ using a machine learning model applied to the RCT's control group and set $b(X) = E[Y^{(0)}|X]$ or $b(X) = -E[Y^{(0)}|X]$ as appropriate so that larger values of $b$ indicate worse outcomes. Additional details are included in Appendix A.

## 3.2 Estimating heterogeneous treatment effects

We estimate heterogeneous treatment effects on each dataset using a doubly-robust estimator (Kennedy, 2023a). The DR estimator splits the data into separate folds. For each fold, we estimate models for both the expected outcome and the treatment variable (estimating the latter even when the propensity scores are known can increase statistical efficiency Hirano et al. (2000)). Let $\hat{\mu}(X, A)$ be the estimated mean outcome for an individual with covariates $X$ and treatment assignment $A$, and $\hat{\pi}(X)$ be the estimated propensity score. For each individual in the held-out data for the fold, we estimate their *pseudo-outcomes*, defined as

$$\chi_i(A) = \hat{\mu}(X_i, A) + \frac{1[W_i = A](Y_i - \hat{\mu}(X_i, A))}{A\hat{\pi}(X_i) + (1 - A)(1 - \hat{\pi}(X_i))}.$$

If at least one of $\hat{\mu}$ or $\hat{\pi}$ is correctly specified, $\chi_i(A)$ has expectation (over the random treatment assignment) equal to $Y_i^{(A)}$, which allows us to use it as a proxy for the unobserved outcomes in evaluating counterfactual evaluation policies.

## 3.3 Exploring treatment effect heterogeneity

Our first analysis tests one potential rationale for risk-based targeting strategies: the hypothesis that individuals with greater baseline risk will also tend to have greater treatment effects. We frame this as estimating $E[Y^{(1)} - Y^{(0)}|b(X)]$, a conditional average treatment effect just with respect to value of the risk score $b$. We follow the doubly-robust approach to estimating CATEs, where the pseudo-outcome difference $\chi_i(1) - \chi_i(0)$ is regressed on the covariates of interest (Kennedy, 2023a). Because our covariate of interest, $b$, is one-dimensional, we use a kernel regression method to estimate the CATE as a generic smooth function (see Appendix A.3).

In later analyses, we compare the welfare gains of potential targeting policies. Here, we also use a doubly-robust estimator. Consider a hypothetical policy that assigns treatments $A(X) \in \{0, 1\}$ as a function of individuals' features $X$. The mean outcome under policy $A$, $\mathbb{E}[Y^{(A(X))}]$, can be decomposed as

$$E[Y^{(A(X))}] = \mathbb{E}[Y^{(0)}] + \Pr(A(X) = 1)\mathbb{E}[Y^{(1)} - Y^{(0)}|A(X) = 1].$$

The term $\mathbb{E}[Y^{(1)} - Y^{(0)}|A(X) = 1]$ represents the treatment effect on the treated population and captures how much the policy improves over no treatment. For policies with the same budget (equal $\Pr(A(X) = 1)$), only this term varies and so we assess allocation policies by their expected treatment-on-the-treated. Following standard doubly-robust estimators for (group) average treatment effects Kennedy (2023b), we empirically estimate this quantity on a separate test set as

$$\frac{1}{\sum_{j=1}^{n} A(X_j)} \sum_{j=1}^{n} A(X_j)(\chi_j(1) - \chi_j(0)). \tag{2}$$

## 3.4 Introducing Confounding

Our next analysis aims to simulate conditions where we do not have access to perfectly conducted randomized controlled trials for our problem, in order to compare risk-based targeting to a plausible alternative in real world settings: targeting according to observational, and potentially biased, estimates of the CATE.

We introduce varying levels of confounding to the RCTs that we study. We do this by simulating adverse selection into treatment, where units are more likely to be observed if the estimated individual-level treatment effects deviate from the mean. Specifically, we generate the biased "observational" dataset by removing data in a systematic manner. This process, inspired by (Kallus & Zhou, 2021), is controlled by a parameter $k$ giving the fraction of data removed, with higher $k$ corresponding to more biased estimates. From the treated units, we remove the examples that lie in the top $k\%$ percent when ordered in descending order of $(\chi_i(1) - \chi_i(0))$ (assuming treatment effect is positive) while for the untreated units, we remove the examples that lie in the bottom $k\%$ of examples when ordered in descending order of $(\chi_i(1) - \chi_i(0))$. In simpler terms, for treated units, we remove examples for which the treatment 'went well'(most positive), while for untreated units, we remove examples for which the lack of treatment did not go well(least positive).

### 3.5 FAMILIES OF WELFARE FUNCTIONS

In order to simulate policymakers with varying preferences for who to treat, we compare risk-based targeting to treatment-effect based targeting with respect to two paradigmatic classes of social welfare functions. A social welfare function maps a vector of individual utilities $\mathbf{u}$ to the policymaker's overall utility for the allocation. Both functions we study are commonly used instances of the class of weighted power mean functions, which contains all social welfare functions satisfying desirable axiomatic properties (Pardeshi et al., 2024). The two functions we investigate here are **_Weighted Utilitarian Welfare_**, given by $M(\mathbf{u}; \mathbf{w}) = \sum_{i=1}^{d} w_i u_i$ and **_Nash Welfare_**, given by $M(\mathbf{u}) = \prod_{i=1}^{d} u_i$.

We consider utilitarian welfare with two sets of weights. First, the uniform weights $w_i = 1 \forall i \in [n]$. Second, $w_i = \frac{n \cdot e^{\alpha b'(X_i)}}{\sum_{j=1}^{n} e^{\alpha b'(X_j)}}$ where $\alpha$ is a hyperparameter and $b'(X_i)$ is represents the percentile score of the baseline risk for the $i^{\text{th}}$ example, with the example with highest baseline risk having score 1 and the example with lowest baseline risk having score 0. This assigns greater weight to individuals with high values of baseline risk for high values of $\alpha$, thereby simulating a scenario where a policymaker might value treating treating these "high risk" individuals. $\alpha$ can be interpreted as $2 \log \left( \frac{w_{75}}{w_{25}} \right)$ where $w_{75}$ is the weight given to the $75^{\text{th}}$ percentile example by baseline risk and $w_{25}$ is the weight given to the $25^{\text{th}}$ percentile example by baseline risk.

The Nash social welfare function has commonly been used to achieve a balance between maximizing total welfare (utilitarian) and ensuring equitable distribution (egalitarian) Caragiannis et al. (2019); Charkhgard et al. (2020). We consider unweighted Nash welfare ($w_i = 1 \forall i$). In order to avoid the complications of utility when using an unweighted Nash welfare function, we scale up the estimated utilities for each example to a minimum value of 1. Note that the Nash welfare can be equivalently formulated in log space Caragiannis et al. (2019) as $\frac{1}{n} \sum_{i=1}^{n} \log u_i$. When each individual's utility under an allocation policy corresponds to their realized outcome $Y_i^{(A(X_i))}$ (e.g., their income after the interventional period), we compare policies exactly as outlined in Equation 2, but with $Y^{(A(X))}$ replaced by $\log Y^{(A(X))}$, estimated by replicating the same procedure after taking logs of all outcome variables.

## 4 RESULTS

Figure 1 shows how treatment effect varies as a function of baseline risk for each of the 5 datasets we study, with 95% confidence intervals shaded around the estimated treatment effects. These intervals are pointwise Wald-type confidence intervals (Kennedy et al., 2019). The estimated relationship between baseline risk and treatment effect is variable across domains. In most domains, the point estimate shows a general upward trend, indicating that individuals at greater risk benefit more (on average) from treatment. However, in the NSW domain, the point estimate is essentially flat. In addition, the confidence intervals are wide for all domains and there is very little statistically significant significant evidence in favor of high-risk individuals benefiting more. Wide confidence intervals reflect that there is significant variance in the pseudo-outcomes estimated for different individuals at the same level of baseline risk. That is, there is a great deal of variance in our estimated treatment effects that is not explained by baseline risk. However, we do expect that risk-based targeting should, in most domains, perform better than a random allocation, since the point estimates generally show larger average effects at higher baseline risk.

Next, Figure 2 shows the the average utility of the three targeting policies – risk-based, treatment effect-based, and random – on each of the datasets. The $x$ axis on each plot represents the degree of confounding synthetically introduced, as discussed above. At the $k = 0$ point ($x = 0$) on the axis, no confounding is present, representing the ideal scenario as in an RCT. For risk-based targeting, we observe a tendency towards improvement over random targeting, but with confidence intervals overlapping in all domains. For treatment-effect based targeting, we observe at most mixed success. On the STAR dataset, we see statistically significant evidence that targeting on treatment effects outperforms risk-based targeting when no confounding is present, with the point estimate trending downwards and eventually drawing even with risk-based targeting as confounding increases. On the NSW dataset, we see a similar trend in the point estimates, but with overlapping confidence intervals. One the remaining datasets, treatment effect targeting has strongly overlapping confidence

intervals with risk-based targeting and shows effectively a flat trajectory with respect to the degree of confounding. We conclude that in these settings, either limited data or insufficiently rich features prevent the model from learning complex treatment effect heterogeneity patterns. On the one hand, we conclude that practitioners should be wary about whether the kind of data generated by many RCTs in practice suffices to learn strong predictors of heterogeneous treatment effects. On the other hand, we find no evidence that the presence of confounding makes matters worse unless there is a meaningful amount of signal present to start with. Appendix A contains results for alternate welfare functions, with similar results.

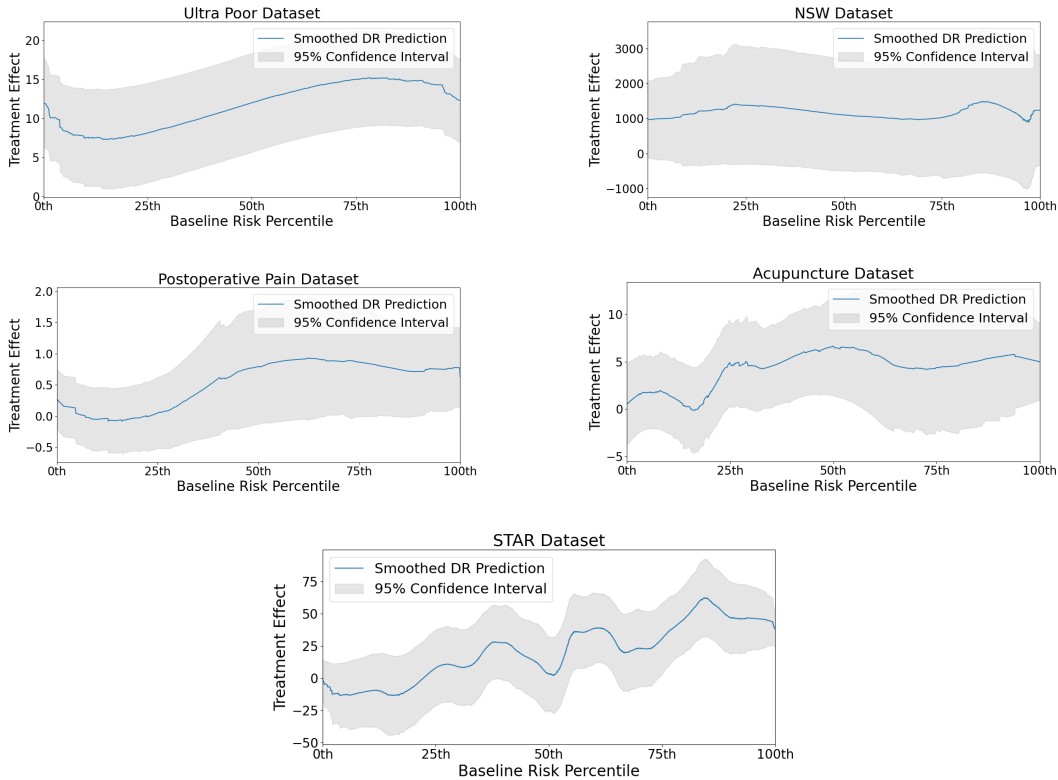

Figure 1: Observing treatment effect heterogeneity across different settings by plotting treatment effect against baseline risk for each of our 5 datasets. We observe a unique trend for each dataset, indicating a lack of a consistent well-defined relation between the two quantities

In order to isolate the impact of confounding in a setting where heterogeneous treatment effects would otherwise be predictable, we now conduct a semi-synthetic analysis that simulates a targeting model with stronger signal. In particular, we simulate a treatment effect-based targeting policy that ranks individuals by their pseudo-outcome directly, instead of the predicted pseudo-outcome using the covariates. This incorporates information on the realized outcome of the individual and so is not attainable in practice. However, it mimics a setting where the second-stage regression model has strong performance in the no-confounding setting. Differences between the two settings are detailed in A.1. The results, shown in Figure 3, demonstrate that when treatment effects can be accurately estimated ($k = 0$, no confounding), targeting based on treatment effects always produces significantly higher utilitarian welfare. This indicates that when a policymaker seeks only to maximize aggregate benefit and can credibly estimate treatment effects, the gains from causal targeting are substantial.

As the level of confounding bias in treatment effect targeting increases (increased $k$), its effectiveness decreases. However, when the policymaker has utilitarian preferences, targeting based on biased treatment effect estimates still performs at least as well as risk-based targeting (and typically better) across all datasets, even for relatively severe levels of confounding. This indicates that using even relatively biased observational data to learn treatment rules is likely superior when the policymaker's goal is just to maximize aggregate gain.

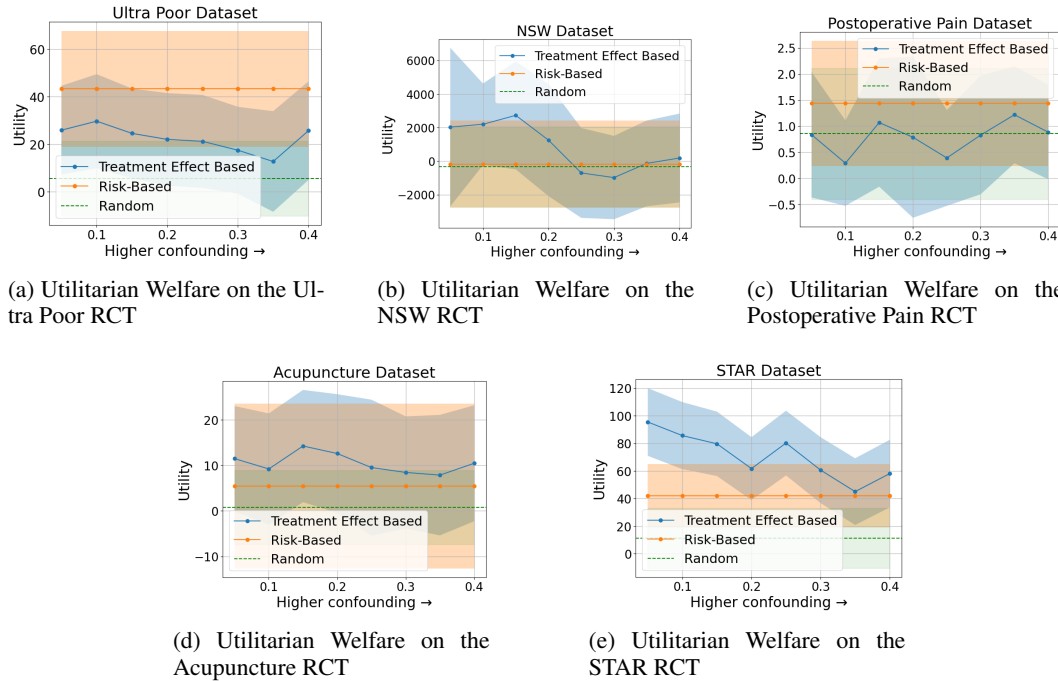

(a) Utilitarian Welfare on the Ultra Poor RCT

(b) Utilitarian Welfare on the NSW RCT

(c) Utilitarian Welfare on the Postoperative Pain RCT

(d) Utilitarian Welfare on the Acupuncture RCT

(e) Utilitarian Welfare on the STAR RCT

Figure 2: Comparison of risk-based targeting to biased treatment effect-based targeting by plotting the benefit offered by each policy against the amount of data systematically removed from the RCT to introduce confounding

The second column of Figure 3 shows an alternative set of preferences, where the policymaker has a utilitarian welfare function with higher welfare weights on individuals with higher baseline risk (for these plots, $\alpha$ is $2 \log 2$, so the ratio of the weight placed on individuals in the 75th percentile of risk to the 25th percentile is 2). This welfare function decreases the gap between treatment effect and risk based targeting as individuals with higher risk now have more weight associated with them. However, targeting on (biased) treatment effects is still preferable to risk-based targeting across all datasets in Figure 3. Similarly, under the Nash social welfare function (third column of Figure 3), we observe some differences across our chosen settings but the general trend remains the same: as we increase confounding (increase the percentage of data we systematically remove), the benefit we accrue by following a treatment assignment policy based on biased treatment effect values decreases. At high levels of confounding, risk-based targeting accrues higher utility for a policymakers with a Nash social welfare function in the Ultra Poor setting in 3. However, across all other datasets, the policymaker prefers to target based on treatment effects even at high levels of confounding.

This motivates us to ask: how much greater must the policymaker weight high-risk individuals in order to prefer risk-based targeting? In Table 1 we give the minimum value of $\alpha$ at which risk-based targeting finally outperformed treatment effect based-targeting at each level of systematic data removal for the real and synthetic settings we consider respectively. We limit $\alpha$ such that the ratio of the 75[th] percentile weight to the 25[th] percentile weight when sorted in descending order is less than 100; otherwise, only a few individuals have non-zero weights and estimating welfare gains becomes impossible. "na" indicates there is no $\alpha$ value (up to our upper bound) at which risk based targeting outperforms targeting based on treatment effects.

In general, we note that the required $\alpha$ values tend to be lower when we increase $k$, which follows directly from the fact that the treatment effect estimates become more biased at higher $k$. We note a dichotomous tendency in the results. In some cases, the value of alpha is 0, where risk-based targeting performs at least as well as treatment-effect based targeting even at $k = 0$). When $\alpha$ takes a non-zero value, it is almost always at least 3 (and typically higher), indicating that the policymaker would need to place approximately 4.5 times more weight on the welfare of an individual at the 75th percentile of baseline risk than an individual at the 25th percentile. We conclude that when treatment effect-based targeting is effective at all, relatively extreme welfare weights are needed to

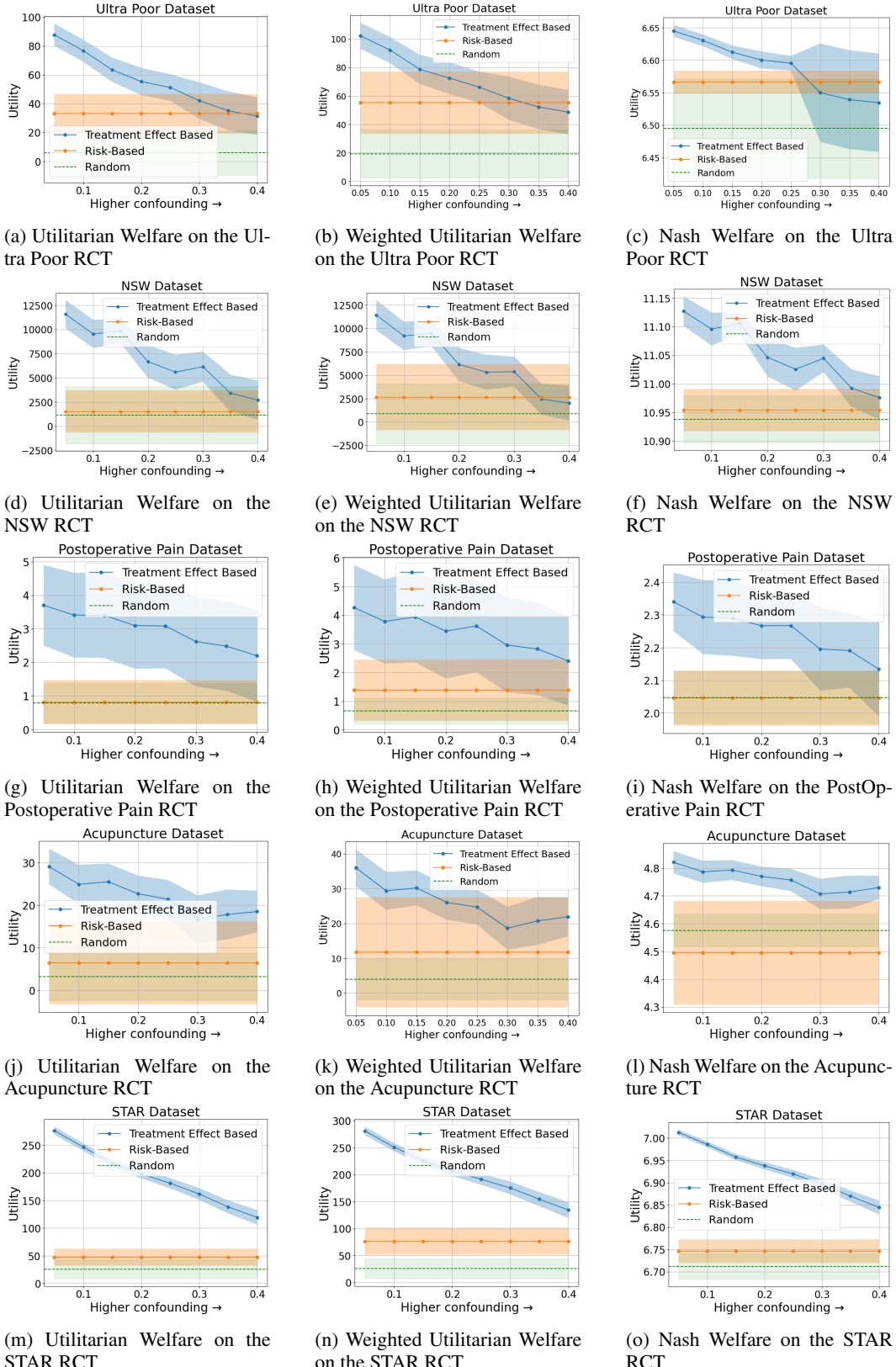

Figure 3: Comparison of risk-based targeting to biased treatment effect-based targeting by plotting the benefit offered by each policy against the amount of data systematically removed from the RCT to introduce confounding but using observed pseudo-outcome data

rationalize the adoption of risk-based targeting instead. Table 2 shows the required $\alpha$ values for the setting when we do not use observed pseudo outcome values for treatment effect estimation.

Table 1: Values of $\alpha$ for different k at which risk-based targeting outperforms treatment effect based targeting. 'na' indicates no such $\alpha$ was found (using observed pseudo-outcome data)

| Dataset | 5% | 10% | 15% | 20% | 25% | 30% | 35% | 40% |
|---|---|---|---|---|---|---|---|---|
| Ultra Poor | na | 8.5 | 5.5 | 4.5 | 3.5 | 2 | 1 | 0 |
| NSW | na | 6.5 | 8.25 | 4.5 | 4.25 | 3.5 | 1.5 | 1 |
| Postoperative Pain | na | 8 | na | 6.5 | na | 5.5 | 6.5 | 4 |
| Acupuncture | na | 7.5 | na | 6.75 | 6 | 3.75 | 4.5 | 5.75 |
| STAR | na | na | na | na | na | na | na | na |

Table 2: Values of $\alpha$ for different k at which risk-based targeting outperforms treatment effect based targeting. 'na' indicates no such $\alpha$ was found

| Dataset | 5% | 10% | 15% | 20% | 25% | 30% | 35% | 40% |
|---|---|---|---|---|---|---|---|---|
| Ultra Poor | 0 | 0 | 0 | 0 | 0 | 0 | 0 | 0 |
| NSW | na | na | na | na | 0 | 0 | na | na |
| Postoperative Pain | 0 | 0 | 0 | 0 | 0 | 0 | 0 | 0 |
| Acupuncture | 3.5 | 3 | na | 9 | 6 | 7 | 7 | 3.5 |
| STAR | 5 | 3 | 3.5 | 2 | 3.5 | 2 | 0.5 | 1.5 |

Collectively, our results reveal two key insights. First, in many real-world settings, limited data makes it difficult to learn reliable mappings from features to treatment effects, which can severely limit the effectiveness of treatment effect based targeting in practice. However, our analysis using directly observed pseudo outcomes, as well as real-data results in the settings with stronger signal, both demonstrate that simple reliability of learning heterogeneous treatment effects is the key bottleneck: if treatment effects could be estimated more reliably (through larger datasets or better modeling approaches), then a policy maker is almost always better of using them for targeting, regardless of both the potential for confounding and inequality-averse preferences.

## 5 CONCLUSION

This paper presents a systematic comparison between two of the most popular treatment assignment policies in use by policymakers today: risk-based targeting and treatment effect based targeting. We find that risk-based targeting tends to produce higher welfare than a uniformly random allocation, confirming some of the intuition behind its widespread use by practitioners. However, our analysis reveals both important practical limitations and significant potential benefits of treatment effect based targeting. The key practical challenge is that real-world datasets can contain insufficient data to learn reliable mappings from features to treatment effects, making it difficult to accurately predict who will benefit most from treatment. This limitation helps explain why risk-based targeting, which requires learning simpler relationships, remains popular in practice.

At the same time, our analysis which simulates a higher-accuracy treatment effect estimator shows the substantial potential benefits of treatment effect based targeting if these estimation challenges could be overcome. Even when accounting for confounding in treatment effect estimates and egalitarian preferences for assisting high-risk individuals, treatment effect based targeting consistently produces better outcomes unless policymakers have extremely strong preferences for helping higher-risk individuals regardless of benefit. One limitation is that our investigation assumes an essentially consequentialist perspective, where the policymaker's goal is to improve individuals' welfare as defined by their outcome. If policymakers have non-consequentialist preferences, targeting directly on a measure of vulnerability may be more appropriate.

Looking forward, we nevertheless find that there would be substantial gains to investing in data and modeling approaches that enabled effective causal targeting (at least with respect to the normative goals we assess). Our results also suggest that enabling strong outcome prediction is worth accepting some risk of bias, potentially arguing in favor of increased use of observational administrative data.

**Acknowledgement:** This work was supported in part by the AI2050 program at Schmidt Futures (Grant G-22-64474).

**Reproducibility Statement:** The supplementary material includes code including data preprocessing and experimentation for each of the datasets. We also detail our procedures in the Appendix A (dataset details) and in Section 4 (step by step experimental procedure).

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

# A  APPENDIX

## A.1  EXPERIMENT DETAILS

**Real Setting:**

We divide the RCT data into two splits such that one split is used for training nuisance functions and the other split is used entirely for evaluation.

Following the DR estimator framework in (Kennedy, 2023a), we divide the first split into two halves and and train a set of outcome and propensity nuisance functions on each half. We then perform cross-fitting where the nuisance functions trained on one half are used to estimate pseudo outcomes for the other half. Finally, we train a random forest regressor strictly between the estimated pseudo outcome differences and a unit's features. This random forest model is applied to units in the evaluation set to obtain treatment effect estimates for the evaluation set. The ground truth treatment effects for the evaluation set are obtained by using information about the observed pseudo outcome and taking the mean of the estimate provided by each set of nuisance functions.

**Semi-synthetic Setting:**

We divide the RCT into two splits such that we use each split to obtain treatment effect estimates for the other split and make maximal use of available data. Again, we use the DR estimator from (Kennedy, 2023a) and use information about the observed pseudo outcome in estimating treatment effects (both ground truth and subsequent biased treatment effect estimates from confounded data).

## A.2  DATASETS

- Targeting the Ultra Poor (TUP) in India ((Banerjee et al., 2021)): This RCT was conducted to study the long-term effects of providing large one-time capital grants to low income-families and observing how family income and overall consumption changed over a period of 7 years. We consider a family's total expenditure as the outcome, which is positively affected by treatment. We filter the dataset before use by removing null values and performing feature selection to limit the number of covariates. The dataset consists of 796 examples post filtering. We quantify baseline risk $b(X)$ as an estimate of baseline expenditure $E[Y(0)|X]$ from a machine learning model, with low values of $E[Y(0)|X]$ corresponding to high baseline risk and vice versa. This follows the hypothesis that households with low expenditure at baseline will benefit most from the treatment.

  While constructing a doubly robust estimator to estimate pseudo outcomes for this dataset, we found the estimated propensity scores to be very high/low for certain examples, which would consequently scale pseudo outcome estimates to unusually large values. Therefore, we manually set propensity scores uniformly according to the treated:untreated ratio in the RCT.

- NSW (National Supported Work demonstration) Dataset ((Dehejia & Wahba, 1999; 2002; LaLonde, 1986): This is a popular causal inference dataset that was used to estimate the impact of the National Supported Work Demonstration, a job training program, on beneficiaries' income in 1978. The covariates include demographic variables like age, race, marital status and academic background, along with the benficiary's income in 1975 prior to the experiment as a baseline. The dataset consists of 722 examples (297 treated and 425 control). Here too, we use an estimate of an individuals baseline income as a measure of risk, following the hypothesis that low income individuals will benefit more from the treatment.

- Tennessee's Student Teacher Achievement Ratio (STAR) project ((Achilles et al., 2008)): The Tennessee State Department of Education conducted a comprehensive four-year study called the Student/Teacher Achievement Ratio (STAR) to examine the effects of class size on student performance. This research, backed by the Tennessee General Assembly, involved 11601 students across 79 schools. The study design included three different classroom configurations:

  - Small classes with 13-17 students per teacher
  - Regular classes with 22-25 students per teacher

– Regular classes with 22-25 students plus a full-time teacher's aide

To ensure unbiased results, both students and teachers were randomly assigned to these different classroom types. The experiment began when the participants entered kindergarten and continued through their third-grade year, allowing for a longitudinal analysis of the impact of class size on educational outcomes. In this paper, we only focus on the first two types of classes mentioned above, so as to stay consistent with treatment value being binary in other RCTs. This large-scale research project aimed to provide empirical evidence on the relationship between class size and student achievement. Again, we filter the dataset before use by removing null values and performing feature selection to limit the number of covariates. We focus on students in kindergarten and a cumulative measure of their scores on various tests as the outcome under consideration. The filtered dataset consists of 3712 students examples. Since we do not have the students' test scores at baseline, we train a random forest model on rows corresponding to students who did not receive the treatment with their test scores at endline being the outcome variable. The prediction offered by this model for every student is then used as a proxy for their baseline test scores and the negative of this value is used as baseline risk. This follows the general hypothesis that students with low test scores need the treatment more.

- Postoperative Pain Dataset: Patients undergoing operations like tracheal intubations often experience throat pain following treatment(Mchardy & Chung, 1999). This RCT was conducted to test the efficacy of gargling with licorice solution prior to endotracheal intubation on reducing postoperative sore throat, which is a common side-effect of the procedure. The investigation involved 236 adult participants scheduled for elective thoracic surgeries necessitating the use of double-lumen endotracheal tubes. The outcome we focus on is a patient's throat pain 4 hours after surgery, measured on a discrete Likert scale from 0 to 7. Additional covariates include a patient's gender, BMI, age, Mallampati score, preoperative pain, surgery size and smoking status. Here, the effect of the treatment is to reduce the amount of throat pain, hence the treatment effect is negative. In order to maintain consistency with other plots, we present results with the sign for treatment effect reversed. Since we do not have a measured value for throat pain at baseline, we again train a random forest model on rows corresponding to patients who did not receive the treatment with their throat pain at endline being the outcome variable. The prediction offered by this model for every patient is then used as a proxy for their baseline throat pain and consequently as the baseline risk. This follows the intuition that patients with more severe throat pain require the treatment more than their co-patients.

- Acupuncture Dataset: This RCT aimed to determine the effect of acupuncture therapy on headache severity in patients with chronic headaches. These measures were assessed at randomization, 3 months post-randomization, and 1 year post-randomization. We focus on headache severity 1 year post-randomization. Headache severity is measured on a discrete Likert scale from 0 to 5. The dataset consists of data from 401 patients with covariates including patient age, sex, chronicity(number of years of headache severity) and whether the headaches were diagnosed as migraines or not. Here again, the effect of the treatment is to reduce the severity of headaches, hence the treatment effect is negative. In order to maintain consistency with other plots, we present results with the sign for treatment effect reversed. We estimate headache severity at baseline $E(Y(0)|X)$ using a machine learning model and use it as a proxy for baseline risk, following the intuition that patients with more severe headaches need the treatment more.

## A.3 KERNEL SMOOTHING

Given the kernel function $K(u) = \exp(-\frac{1}{2}u^2)$, the CATE estimate at $b(X_i)$ is given by:

$$\hat{\tau}(b(X_i)) = \frac{\sum_{j=1}^{n} K\left(\frac{b(X_j)-b(X_i)}{\sigma_i}\right)\hat{\tau}_j}{\sum_{j=1}^{n} K\left(\frac{b(X_j)-b(X_i)}{\sigma_i}\right)} \tag{3}$$

where $\hat{\tau}_j$ is the estimated difference in pseudo outcomes, for unit $j$, $\chi_j(1) - \chi_j(0)$, as determined by the doubly robust estimator, and $\sigma_i$ is the adaptive bandwidth calculated as:

$$\sigma_i = \frac{1}{2}(b(X_{i+100}) - b(X_{i-99})) \tag{4}$$

for a window of 200 data points centered at $i$. The confidence intervals are computed using a weighted variance estimate:

$$\text{CI} = \hat{\tau}(b(X_i)) \pm 1.96 \sqrt{\frac{\sum_{j=1}^{n} K(\frac{b(X_j)-b(X_i)}{\sigma_i})(\hat{\tau}_j - \hat{\tau}(b(X_i)))^2}{(\sum_{j=1}^{n} K(\frac{b(X_j)-b(X_i)}{\sigma_i}))^2}} \tag{5}$$

This approach allows us to capture the heterogeneity in treatment effects across different levels of baseline risk while accounting for the varying density of data points.

## A.4 ADDITIONAL PLOTS

### A.4.1 EVALUATING BASED ON WEIGHTED UTILITARIAN WELFARE AND NASH SOCIAL WELFARE

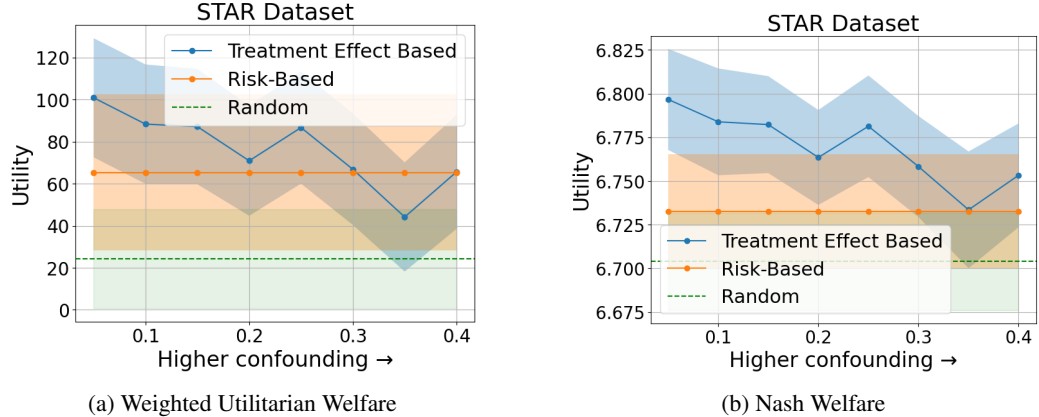

(a) Weighted Utilitarian Welfare

(b) Nash Welfare

Figure 4: Comparing risk-based targeting to biased treatment effect-based targeting on weighted utilitarian welfare and Nash social welfare for the STAR RCT.

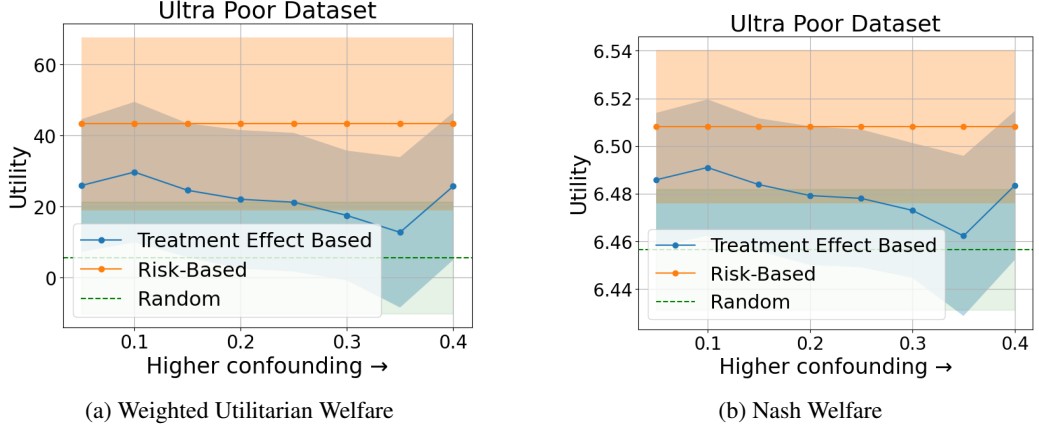

(a) Weighted Utilitarian Welfare

(b) Nash Welfare

Figure 5: Comparing risk-based targeting to biased treatment effect-based targeting on weighted utilitarian welfare and Nash social welfare for the Ultra Poor RCT.

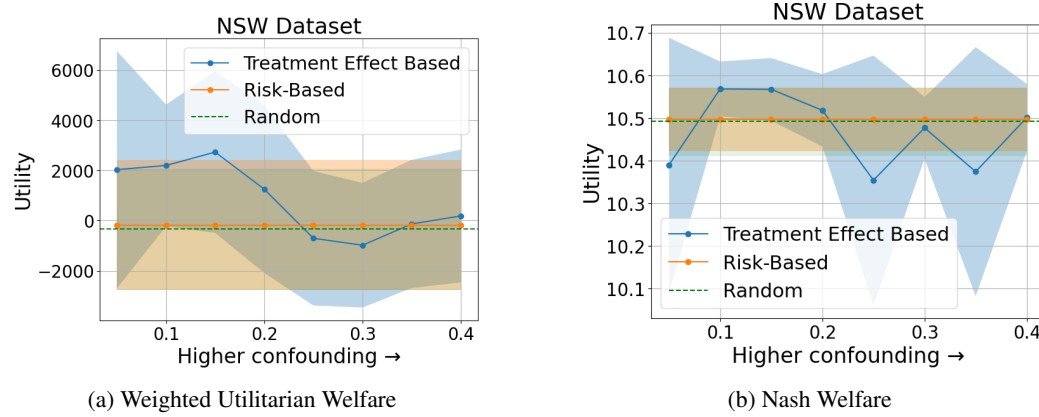

(a) Weighted Utilitarian Welfare

(b) Nash Welfare

Figure 6: Comparing risk-based targeting to biased treatment effect-based targeting on weighted utilitarian welfare and Nash social welfare for the NSW RCT.

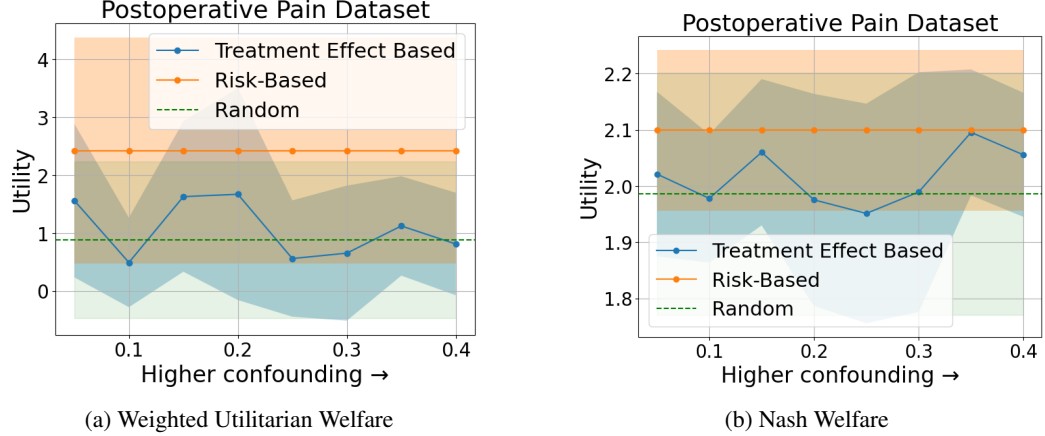

(a) Weighted Utilitarian Welfare

(b) Nash Welfare

Figure 7: Comparing risk-based targeting to biased treatment effect-based targeting on weighted utilitarian welfare and Nash social welfare for the Postoperative Pain RCT.

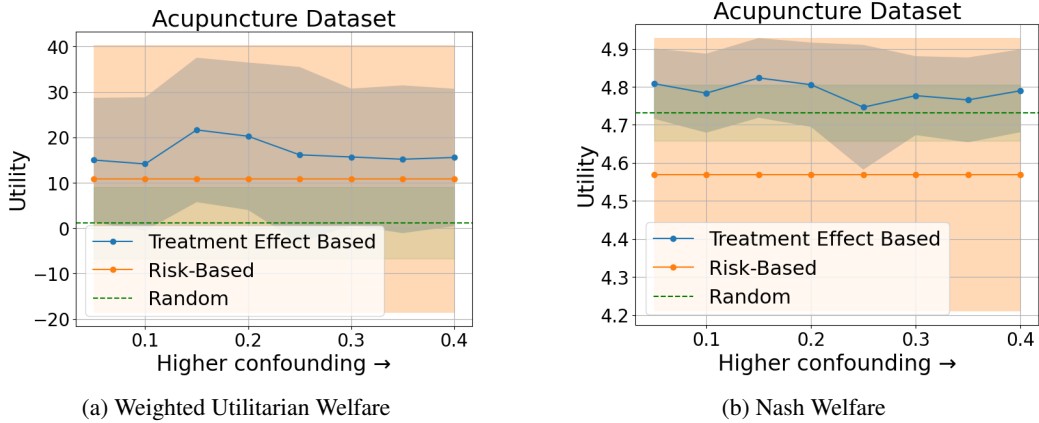

(a) Weighted Utilitarian Welfare

(b) Nash Welfare

Figure 8: Comparing risk-based targeting to biased treatment effect-based targeting on weighted utilitarian welfare and Nash social welfare for the Acupuncture RCT.

A.4.2  EVALUATING BASED ON UTILITARIAN WELFARE WITH BUDGET = 30% AND 40% OF
       THE POPULATION BUT USING OBSERVED PSEUDO OUTCOME INFORMATION

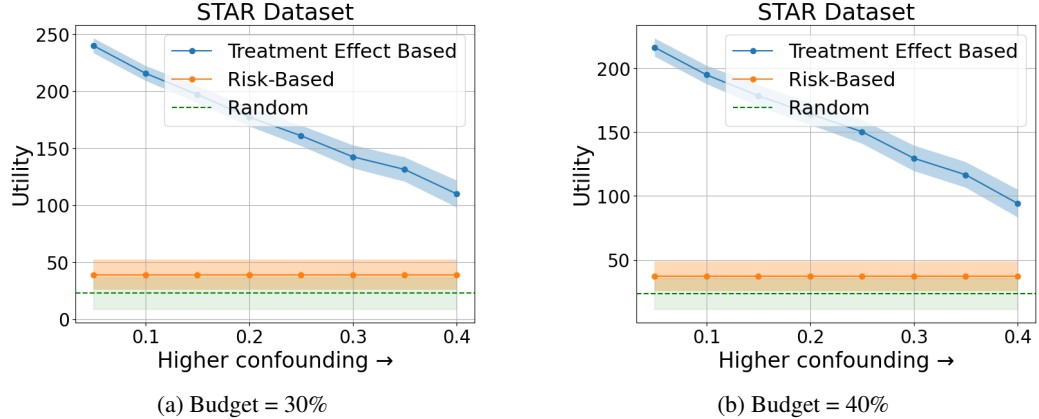

Figure 9: Comparison of risk-based assignment to biased treatment effect based assignment for the STAR dataset, with fixed budget of 30% and 40% of the population respectively.

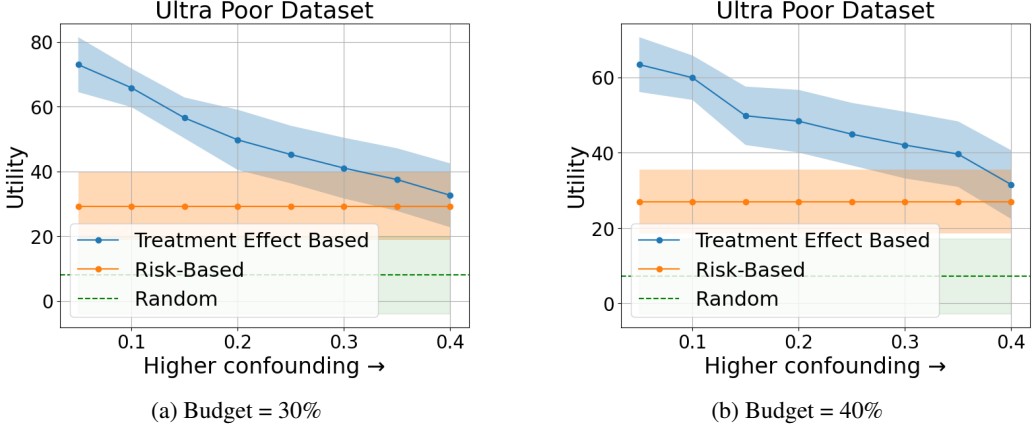

Figure 10: Comparison of risk-based assignment to biased treatment effect based assignment for the Ultra Poor dataset, with fixed budget of 30% and 40% of the population respectively.

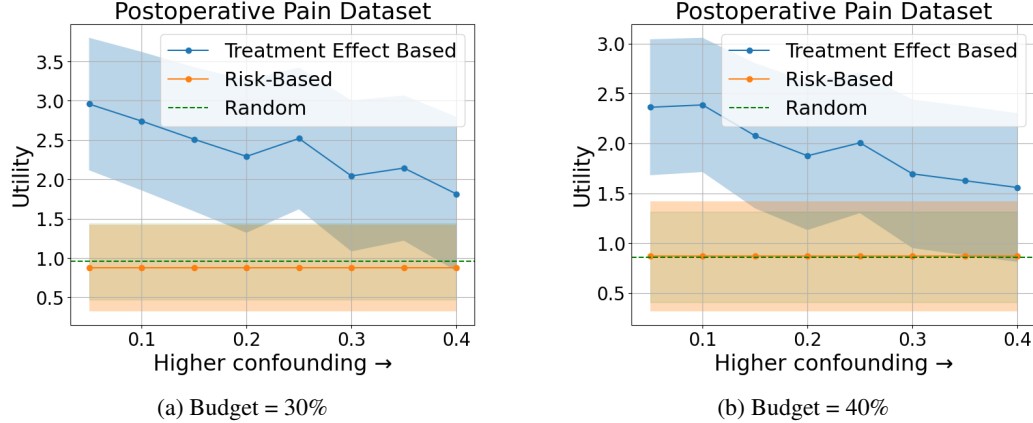

Figure 11: Comparison of risk-based assignment to biased treatment effect based assignment for the Postoperative Pain dataset, with fixed budget of 30% and 40% of the population respectively.

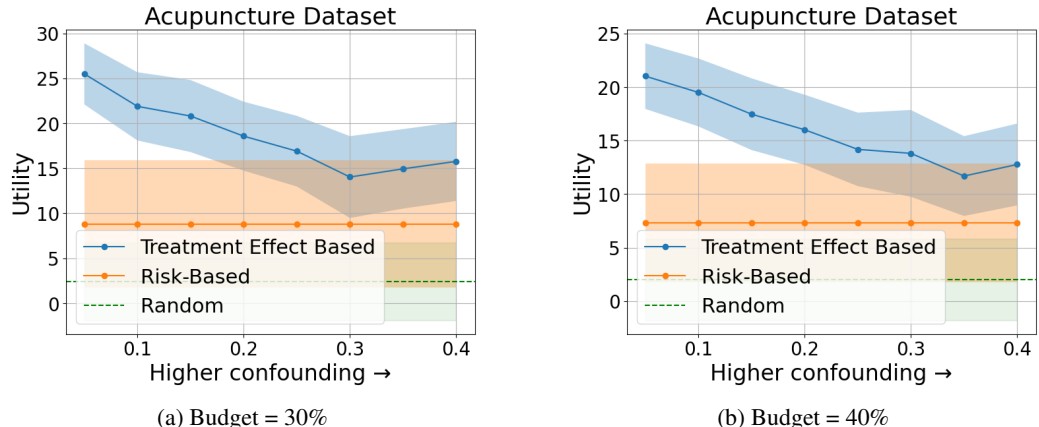

Figure 12: Comparison of risk-based assignment to biased treatment effect based assignment for the Acupuncture dataset, with fixed budget of 30% and 40% of the population respectively.

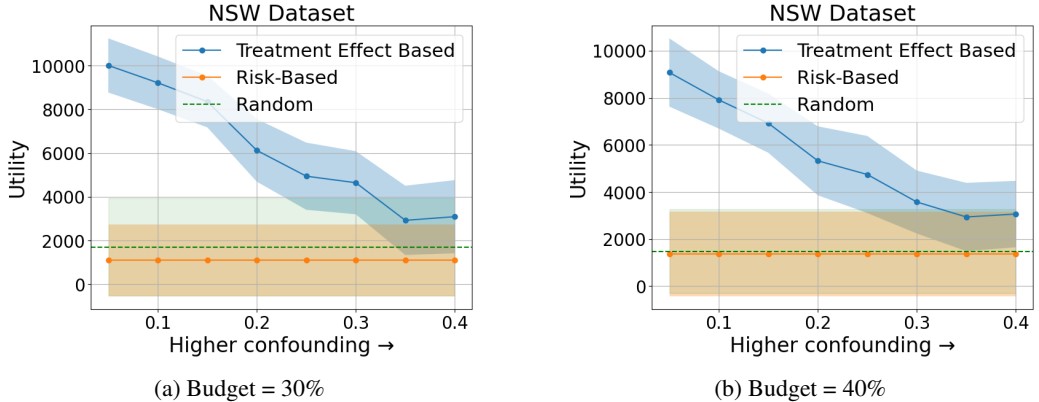

Figure 13: Comparison of risk-based assignment to biased treatment effect based assignment for the NSW dataset, with fixed budget of 30% and 40% of the population respectively.

### A.4.3 EVALUATING BASED ON UTILITARIAN WELFARE WITH BUDGET = 30% AND 40% OF THE POPULATION

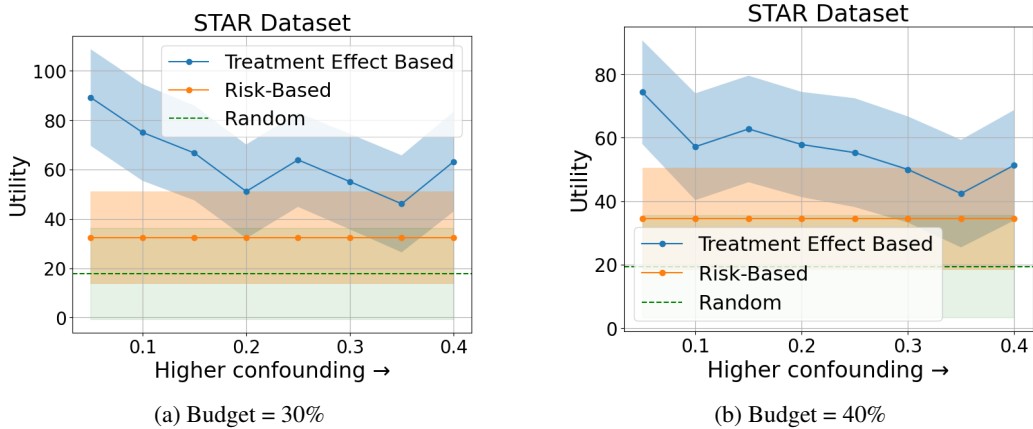

(a) Budget = 30%  (b) Budget = 40%

Figure 14: Comparison of risk-based assignment to biased treatment effect based assignment for the STAR dataset, with fixed budget of 30% and 40% of the population respectively.

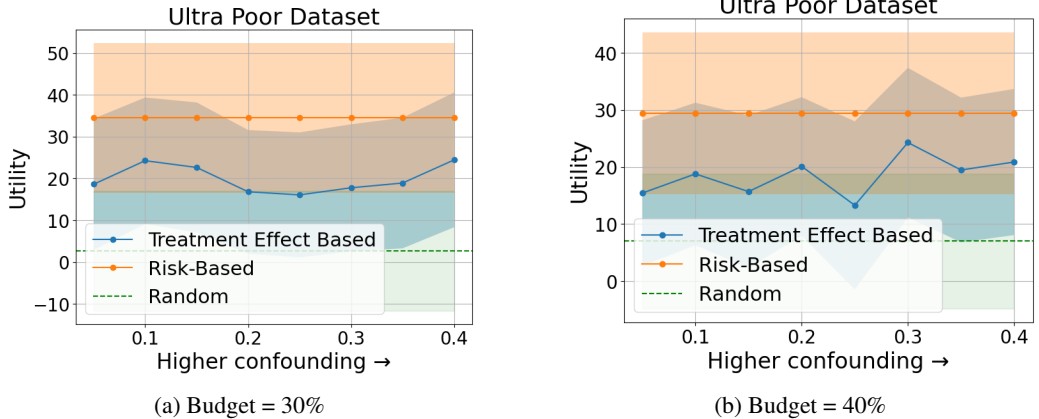

(a) Budget = 30%  (b) Budget = 40%

Figure 15: Comparison of risk-based assignment to biased treatment effect based assignment for the Ultra Poor dataset, with fixed budget of 30% and 40% of the population respectively.

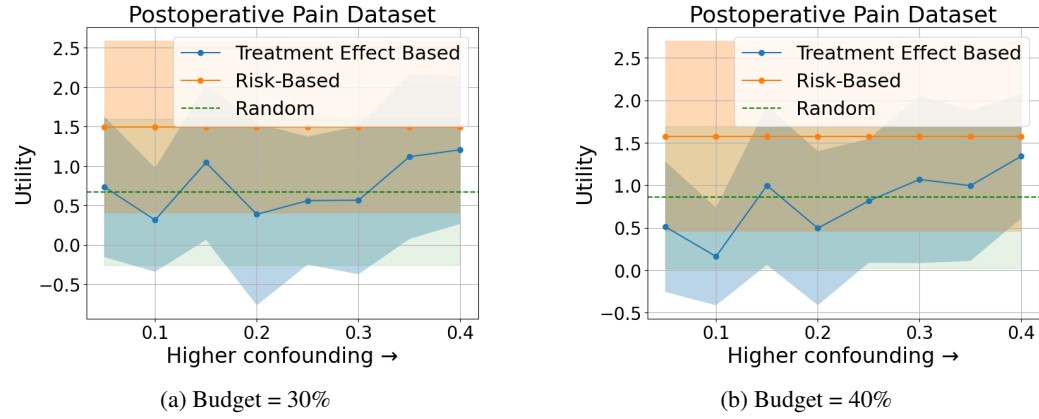

Figure 16: Comparison of risk-based assignment to biased treatment effect based assignment for the Postoperative Pain dataset, with fixed budget of 30% and 40% of the population respectively.

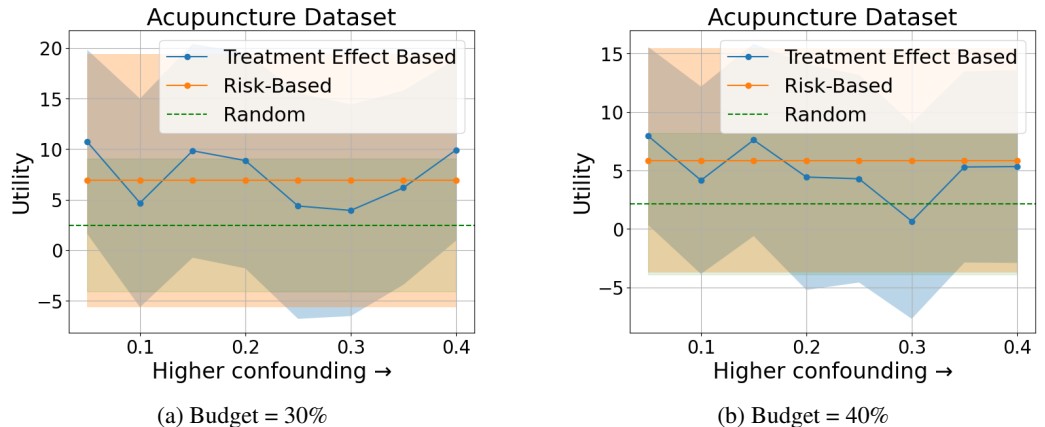

Figure 17: Comparison of risk-based assignment to biased treatment effect based assignment for the Acupuncture dataset, with fixed budget of 30% and 40% of the population respectively.

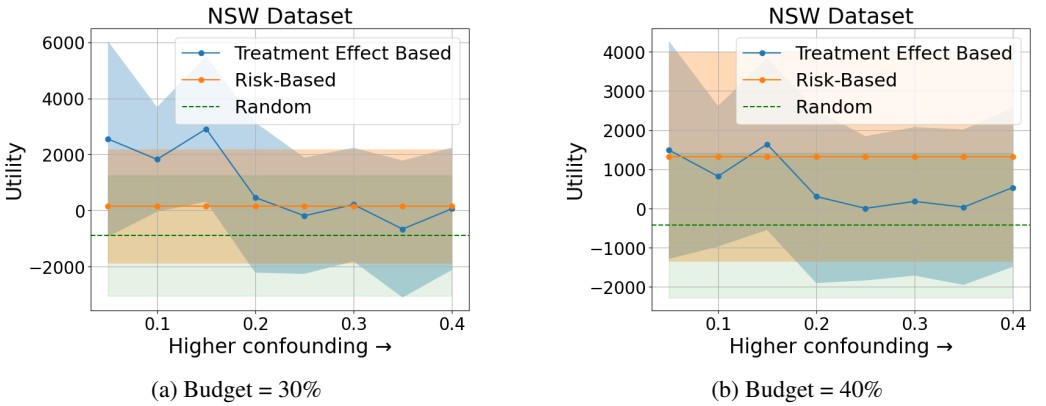

Figure 18: Comparison of risk-based assignment to biased treatment effect based assignment for the NSW dataset, with fixed budget of 30% and 40% of the population respectively.

