# OpenReview forum: "Comparing Targeting Strategies for Maximizing Social Welfare with Limited Resources"
_ICLR.cc/2025/Conference — ICLR 2025 Poster_

### Official Review · Reviewer_WHnh · 2024-11-01

**Soundness:** 4
**Presentation:** 4
**Contribution:** 4
**Rating:** 10
**Confidence:** 4

**Summary:**

This paper studies a common risk-based approach to treatment allocation applied to a diverse range of treatments with randomized controlled trial data available. The authors find that allocating treatment according to risk (ie, bad outcomes under the no-treat baseline) produces worse outcomes than allocating according to conditional average treatment effect estimates, even when CATEs are biased or the decision maker has a preference for treating high-risk individuals.

**Strengths:**

This is a simple, well-executed paper making a very important point. As the authors note, most algorithmic decision making approaches are based on risk or some manipulation of risk in pursuit of fairness goals. This paper demonstrates the problem with that approach across a wide range of interventions.

The authors anticipate the key criticisms of the policy based approach (that RCT data is hard to find and so CATEs may be biased, and that equity preferences might provide a non-utilitarian reason to prefer risk-based targeting), and provide compelling evidence that even under significant confounding or equity preferences one should still prefer allocations based on CATEs.

**Weaknesses:**

The weakest part of this paper is probably the discussion of Fig 1. The authors cite a "unique trend for each dataset" (in CATE conditional on risk score) as evidence that the risk-based policy is flawed. Visually, to me, the trends look pretty similar: they look like there is basically no relationship between CATE and risk for most RCTs studied. This is still evidence for the risk-based policy being flawed (so it doesn't invalidate the claim being made), but I think it's a more accurate way to describe the results. I think Fig 2 is more compelling than Fig 1 anyway, so it probably make sense to lead with that result for the most impactful paper.

**Questions:**

Fig. 2 is great, almost telling the story of the paper without the audience needing to read anything else. What would make it even better is replacing "Percentage of data removed" with something like "Higher values mean more confounding in CATE estimates". Currently readers have to read the methodology to understand the x axis, but a better label would mean they could understand the x axis without knowing exactly how the confounding was introduced, and read the methodology if they wanted more details.

---

> ### Author Response · Authors · 2024-11-21
> **Reply to Reviewer 4**
>
> Thank you very much for the strengths discussed in your review! Regarding the weakness you point out: multiple reviewers pointed out that the interpretation of Figure 1 is ambiguous and suggest various alternative stories consistent with the plot. We agree with the reviewers and thank them for raising this point -- all of the interpretations that the reviewers mention do find some support in Fig 1 due to the wide confidence intervals. However, our main conclusions are driven by Fig 2, where we are able to draw much more statistically precise conclusions. We have edited the text to frame Fig 1 just as suggesting hypotheses about the effects of risk based targeting, which Fig 2 addresses with more precision/statistical significance. Here is the revised text discussing Fig 1:
>
> The estimated relationship between baseline risk and treatment effect is variable across domains. In most domains, the point estimate shows a general upward trend, indicating that individuals at greater risk benefit more (on average) from treatment. However, in the NSW domain, the point estimate is essentially flat. In addition, the confidence intervals are wide for all domains and there is very little statistically significant significant evidence in favor of high-risk individuals benefiting more. Wide confidence intervals reflect that there is significant variance in the pseudo-outcomes estimated for different individuals at the same level of baseline risk. That is, there is a great deal of variance in our estimated treatment effects that is not explained by baseline risk. From these results, we form two hypotheses. First, that risk-based targeting should, in most domains, perform better than a random allocation, since the point estimates generally show larger average effects at higher baseline risk. Second, that there is room to improve on risk-based targeting via strategies that leverage some of the substantial variance in treatment effects that is unexplained by baseline risk. The next section provides more statistically precise tests of these hypotheses by comparing the welfare associated with each targeting policy (a single number, which can be quantified more precisely than the entire curve shown in Figure 1).
>
> We have also edited the introduction and conclusion of the paper to reflect this framing. We also thank you for your suggestion to improve Figure 2: we have incorporated it in the new manuscript.

---

### Official Review · Reviewer_XFvr · 2024-11-02

**Soundness:** 2
**Presentation:** 3
**Contribution:** 3
**Rating:** 6
**Confidence:** 3

**Summary:**

The paper compares ''risk-based targeting'' and ''treatment effect-based targeting'' across five datasets from RCTs in different domains. In risk-based targeting, the planner targets individuals based on their baseline risk, which is the expected outcome in the absence of treatment. This approach, commonly used by practitioners, does not account for causal effects. In contrast, treatment effect-based targeting involves first estimating the Conditional Average Treatment Effect (CATE) and then prioritizing individuals accordingly, often with the help of machine learning. A key concern is that these methods may introduce bias in the presence of unobserved confounding. The paper aims to provide empirically grounded guidance for navigating this tradeoff.

First, the paper investigates the extent to which baseline risk serves as an effective proxy for targeting. To do this, treatment effects are estimated at different levels of baseline risk, revealing a surprisingly complex and not necessarily monotonic relationship between treatment effect and baseline risk. Second, the paper examines the potential additional gains from targeting based on treatment effect. Both utilitarian and Nash welfare are used to compare these mechanisms, showing that targeting based on estimated treatment effect can be up to three times more effective in nearly all cases. This remains true even when confounding is introduced into the CATE estimation and when the policymaker's preference for risk-based targeting is encoded in the social welfare function.

**Strengths:**

This is an important issue, and I’ve personally never found a clear guideline on how best to approach it. I found the study original and of good quality.

The paper does a commendable job of incorporating datasets from diverse domains.

It also simulates the impact of unobserved confounders and considers the possibility of policymaker bias toward risk-based targeting.

**Weaknesses:**

My main concern is the reliability of comparing the estimated welfare from risk-based versus treatment effect-based targeting. The issue with the latter approach, particularly in resource-scarce settings, is that it first estimates CATE, then selects individuals with the highest estimated CATE given the available budget, and finally estimates the welfare of treating these individuals using the same estimated CATE. Even if the CATE is unbiased, this procedure can lead to severely biased welfare estimates in resource-constrained settings.

In Figure 1, the paper provides treatment effect estimates for each baseline risk percentile, along with confidence intervals. What exactly are we looking for here? If the goal is to rule out a monotonic relationship between baseline risk and treatment effect, none of the figures offer statistically significant evidence to support this.

I also think the results should intuitively depend on the budget but this is not discussed in the paper. For instance, if the budget is very small, treatment effect-based allocation can be even more effective in exploiting heterogeneity.

Overall, the writing is strong. I noticed the following minor typos: page 2 (potentially-based -> potentially-biased), page 4 (policy -> policy), page 5, second equation, page 7, Equation 4.5 is referenced which does not exist

**Questions:**

Feel free to further discuss the first two weaknesses mentioned above.

I have seen papers, such as [1], that discuss the complexities of estimating allocation welfare under budget constraints. The issue is that these constraints introduce dependencies between individual estimations, complicating the evaluation process. How is this relevant to your setting?

Shouldn't the results also depend on how many individuals are treated in each scenario? My intuition (supported by studies like [2]) suggests that the relative budget plays a significant role in determining which type of allocation mechanism is optimal.

Since you studied a wide range of settings, did you find any recommendations that vary across these contexts? For instance, are there conditions under which risk-based targeting performs better?

Could you elaborate on the choice of outcome of interest in the TUP dataset? It seems there’s an implicit assumption that individuals who show a larger increase in expenditure are more deserving of intervention. Why should this be the case? In contrast, a risk-based approach might assume that those who would have lower expenditures in the absence of interventions are more deserving. Could this actually be a more appropriate assumption?

[1] Improved Policy Evaluation for Randomized Trials of Algorithmic Resource Allocation
[2] Allocation Requires Prediction Only if Inequality Is Low

---

> ### Author Response · Authors · 2024-11-21
> **Reply to Reviewer 3**
>
> We thank you for your detailed review and constructive feedback. We address the weaknesses mentioned in the review in this comment and the questions in the subsequent comment. We have also edited the paper to incorporate the feedback we received and the edits are marked in red for easy distinction.
>
> Wekanesses:
>
> 1) **"My main concern is..."***
>
> To address this concern, we employ a sample splitting approach, in line with the literature on doubly robust CATE estimation and previous work on policy optimization/comparison. In particular, our strategy is equivalent to the cross-validation strategy used in the experiments section of Athey and Wagner [1] to evaluate learned policies on RCT data. The theoretical basis for this strategy is that, with randomized data, the estimated welfare gains should be unbiased even conditional on the learned outcome model used for targeting (which may be biased) since the propensity score is guaranteed to be well-specified.
> To further address this concern though, we added an additional robustness check in Appendix B, beyond what previous work has done. In this analysis, instead of using a cross-validation approach, we hold out half the dataset only for evaluation so that the estimator used for evaluation was trained on entirely disjoint data from the targeting policy. This substantially reduces the amount of data available for either the learning or evaluation tasks, and so is only possible with the two largest datasets we use. However, we find generally consistent results with our main analysis, where causal targeting performs better than risk-based, even under moderate levels of confounding or inequality-aversion.
> [1] Athey and Wager. Policy Learning with Observational Data. Econometrica 2021.
>
> 2) **"In Figure 1...**:
>
> Answer: Multiple reviewers pointed out that the interpretation of Figure 1 is ambiguous and suggest various alternative stories consistent with the plot. We agree with the reviewers and thank them for raising this point -- all of the interpretations that the reviewers mention do find some support in Fig 1 due to the wide confidence intervals. However, our main conclusions are driven by Fig 2, where we are able to draw much more statistically precise conclusions. We have edited the text to frame Fig 1 just as suggesting hypotheses about the effects of risk based targeting, which Fig 2 addresses with more precision/statistical significance. Here is the revised text discussing Fig 1:
>
> The estimated relationship between baseline risk and treatment effect is variable across domains. In most domains, the point estimate shows a general upward trend, indicating that individuals at greater risk benefit more (on average) from treatment. However, in the NSW domain, the point estimate is essentially flat. In addition, the confidence intervals are wide for all domains and there is very little statistically significant significant evidence in favor of high-risk individuals benefiting more. Wide confidence intervals reflect that there is significant variance in the pseudo-outcomes estimated for different individuals at the same level of baseline risk. That is, there is a great deal of variance in our estimated treatment effects that is not explained by baseline risk. From these results, we form two hypotheses. First, that risk-based targeting should, in most domains, perform better than a random allocation, since the point estimates generally show larger average effects at higher baseline risk. Second, that there is room to improve on risk-based targeting via strategies that leverage some of the substantial variance in treatment effects that is unexplained by baseline risk. The next section provides more statistically precise tests of these hypotheses by comparing the welfare associated with each targeting policy (a single number, which can be quantified more precisely than the entire curve shown in Figure 1).
>
> We have also edited the introduction and conclusion of the paper to reflect this framing.
>
> 3) **"I also think..."**
>
> We provide results for 3 budget levels in the paper: 20% of the total population (main paper), 30% of the total population (appendix A.2), 40% of the total population (appendix A.2). The trends for the latter 2 appear to match the trend we see with a budget of 20%.
>
> We thank you for bringing the typographic errors to our notice. These have been corrected in the updated manuscript.

---

> > ### Author Response · Authors · 2024-11-21
> > **Reply to Reviewer 3**
> >
> > Questions:
> >
> > 1) Regarding **"I have seen..."**
> >
> > [1] addresses dependencies caused by hard constraints in a finite population. As our focus is not on the complications of exactly enforcing hard constraints, our results can be seen as evaluating a policy which treats the desired fraction of individuals in-expectation but makes the actual treatment assignments independently (potentially over- or under-utilizing the budget in any specific time frame).
> >
> > 2) Regarding **"Shouldn't the results..."**
> >
> > We provide results for 3 budget levels in the paper: 20% of the total population (main paper), 30% of the total population (appendix A.2), 40% of the total population (appendix A.2). The trends for the latter 2 appear to match the trend we see with a budget of 20%.
> >
> > 3) Regarding **"Since you studied..."**
> >
> > Answer: Risk-based targeting appears to perform better when there is the combination of a significant amount of confounding in the treatment effect estimates along with very strong egalitarian preferences from the policymaker. However, either factor on its own is generally insufficient.
> >
> > 4) Regarding **"Could you elaborate..."**
> >
> > For the TUP dataset we use expenditure because it is one of the main outcomes selected by the authors of the original RCT [3] and the treatment has a positive average effect on it.
> >
> > Regarding whether the policymaker should aim to improve individuals’ average economic circumstances or whether they should target based on desert (where individuals with low expenditures may deserve aid more): this is a normative question outside the scope of our paper. We assess tradeoffs in a consequentialist framework where policies are evaluated based on the extent to which they improve outcomes, not whether they align with a notion of desert.
> >
> > We have edited the conclusion to acknowledge this:
> > “Our investigation of egalitarian preferences assumes an essentially consequentialist perspective, where the policymaker's goal is to improve individuals' welfare as defined by their outcome. If policymakers have non-consequentialist preferences, for example viewing the assistance of those in need as an inherent good regardless of its effects, targeting directly on a measure of vulnerability may be more appropriate.”
> >
> > *[3] Banerjee, Abhijit, Esther Duflo, Rachel Glennerster, and Cynthia Kinnan. 2015. "The Miracle of Microfinance? Evidence from a Randomized Evaluation." American Economic Journal: Applied Economics, 7 (1): 22–53.*
> >
> > We thank you again for your feedback.

---

> ### Comment · Reviewer_XFvr · 2024-11-21
> **Response to Rebuttal**
>
> Thank you for your response.
>
> I reviewed Appendix B and appreciate the robustness analysis you conducted. However, I am concerned about the stark contrast between Figure 2(d) and Figure 8(a) (or Figure 10). I understand that training the DR estimator on a dedicated split while choosing targeting decisions on a separate split effectively reduces the sample size by half. However, I cannot understand why this would lead to such dramatic changes in the results for the NSW dataset.
>
> I also appreciate the reframing around Figure 1. I agree that this figure can only serve as a suggestive illustration and that the main argument should center on Figure 2.
>
> Overall, I believe the language of the paper has improved during the rebuttal process. That said, I still find the conclusions less definitive than I expected, and the contrast between the results in the main text and Appendix B remains somewhat concerning. I wish I could assign a score of 5.5, but since that's not an option, I will raise my score to 6 :)

---

### Official Review · Reviewer_1WAp · 2024-11-03

**Soundness:** 3
**Presentation:** 2
**Contribution:** 3
**Rating:** 5
**Confidence:** 4

**Summary:**

The paper explores targeting strategies for allocating limited resources to maximize social welfare in areas like
human services, healthcare, and education. Specifically, it compares risk-based targeting, which prioritizes
high-risk individuals, with treatment-effect-based targeting, which uses machine learning models to estimate who
might benefit the most from interventions. Using five real-world randomized controlled trials (RCTs) across diverse
domains, the paper concludes that even biased estimates of treatment effects generally outperform risk-based
targeting. This finding suggests in addition to the widespread reliance on risk-based approaches, policymakers
could incorporate treatment effect estimation when feasible.

**Strengths:**

Scope: The work’s target is evident in its cross-domain empirical evaluation, making it applicable to multiple policy
areas. The paper introduces innovative use of biased treatment effect estimates to assess targeting efficacy.
Quality: Methodologically, the paper is solid, employing credible RCT data and a robust approach to measuring
treatment effect heterogeneity. The use of doubly robust estimation and varying social welfare functions adds
depth to the analysis.

**Weaknesses:**

Grammatical errors: there are notable grammatical errors in spelling and following the formatting of ICLR
instructions.

Scope of Treatment Effect Estimates: The reliance on simulated confounding could be more thoroughly justified;
real-world application requires consideration of domain-specific biases that could vary across different policy areas
Potential Ethical Implications: Since the work could influence resource allocation in sensitive domains, further
discussion on ethical considerations regarding inequality and bias would strengthen its impact and application.

**Questions:**

Modeling Real-World Confounding: Could the authors expand on how their confounding
approach aligns with real-world biases encountered in observational data? A discussion on potential limitations in modeling confounding factors might aid readers in interpreting results for specific applications.

Ethical Implications: How might the authors' conclusions address ethical concerns,
particularly in terms of balancing fairness (people at the most rish) with effectiveness
when treatment effect targeting benefits some groups more than others? Because the paper concludes that a treatment-effect-based method is preferable to a risk-based method, it raises
the natural question of whether individuals at the highest risk should be prioritized, or if those with the greatest
potential treatment outcome should be targeted instead.

Data Quality: Could the authors provide more information on the reliability and source of the
Acupuncture Dataset?

Confidence Interval Estimation: In Equation (5), a biased estimator is used. Could the authors
justify why this choice was made? Was it primarily for computational efficiency, or are there
other reasons for using a biased estimator in this context?

---

> ### Author Response · Authors · 2024-11-21
> **Reply to Reviewer 2**
>
> We thank you for your detailed review and constructive feedback.
>
> **Regarding Question: "Modeling Real-World Confounding...":**
>
> To provide a strong comparison, we simulate a very adversarial mode of confounding, where individuals select into treatment depending on the actual value of their potential outcome. This can be seen as a strengthening of a typical mechanism for confounding, where individuals are selected for treatment based on unobservable characteristics that are merely correlated with their potential outcomes. We edited the paper to explain this motivation.
>
> **Regarding Question: "Ethical Implications...":**
>
> This is an important question. We address it in the paper by evaluating policies under two families of egalitarian welfare functions which place greater weight on the welfare of individuals with greater risk. This provides guidance for policymakers based on the degree of preferences for fairness which is appropriate in a given domain. We have also acknowledged that there could be additional ethical issues by adding the following to the conclusion: “Our investigation of egalitarian preferences assumes an essentially consequentialist perspective, where the policymaker's goal is to improve individuals' welfare as defined by their outcome. If policymakers have non-consequentialist preferences, for example viewing the assistance of those in need as an inherent good regardless of its effects, targeting directly on a measure of vulnerability may be more appropriate.”
>
> **Regarding Question:  "Data Quality...":**
>
> The acupuncture dataset contains data obtained from a study published in the British Medical Journal (a top medical publication): “Acupuncture for chronic headache in primary care: large, pragmatic, randomized trial”, Andrew J Vickers, Rebecca W Rees, Catherine E Zollman, Rob McCarney, Claire M Smith, Nadia Ellis, Peter Fisher and Robbert Van Haselen, BMJ, 2004 (https://pmc.ncbi.nlm.nih.gov/articles/PMC381326/ which has been cited 448 times). It is available at https://www.causeweb.org/tshs/acupuncture/ and was contributed by Dr. Steven C. Grambow, Director, Duke Clinical Research Training Program and Assistant Professor, Duke University.
>
> **Regarding Question: "Confidence Interval Estimation...":**
>
> The estimator is based on an asymptotic normal approximation, which is indeed computationally efficient. Note that, as we emphasize now in the paper, we are not attempting to draw statistically significant conclusions from Figure 1, so exact CIs are less important.
>
> We thank you again for your feedback.

---

> ### Author Response · Authors · 2024-11-25
>
> With the discussion period ending soon, we'd like to check if the above revisions to the paper address your concerns. Particularly since much of the discussion of this paper centers on the interpretation/framing of the results, we're hoping to make the most of the opportunity to improve the paper!

---

> > ### Comment · Reviewer_1WAp · 2024-11-26
> >
> > Thank you for your response.
> >
> > The reasoning and supporting references in your rebuttal have sufficiently addressed all my questions. It would be helpful, however, to include a sensitivity analysis to demonstrate the efficiency of this asymptotic normal approximation estimator when applied to datasets of varying sizes, particularly those targeting the limited population of interest.
> >
> > I acknowledge that the ethical implications are challenging to address comprehensively, but your additional reasoning in the conclusion is valuable. While not essential, it might enhance the discussion to provide a rough intuition on implementing consequentialist and non-consequentialist perspectives, such as a mixed strategy approach.
> >
> > While your study may not address every question of interest to practitioners or policymakers, it nonetheless provides valuable insights into targeting strategies.

---

### Official Review · Reviewer_4SA7 · 2024-11-04

**Soundness:** 3
**Presentation:** 3
**Contribution:** 2
**Rating:** 6
**Confidence:** 4

**Summary:**

The authors compare the utility of targeting interventions based on estimated treatment effects with the utility of targeting based on predicted risk. The former is the method of choice from a utilitarian perspective. But practitioners and policy makers often choose the latter due to its simplicity and in cases where the normative goal is to assist those in greatest need.

**Strengths:**

(1)

The comparison of these two targeting strategies is an important problem. I'm glad to see that the authors study this question. Despite its significance, there hasn't been much reliable insight so far. I'd love to see more work in this direction!  I'm weighing this strongly in my evaluation.

(2)

The paper is very clearly written. The authors narrate a compelling story about the advantages of targeting based on treatment effect estimates, even if they are biased.

**Weaknesses:**

(1)

As compelling as the story is, I find the results less than conclusive. Looking at Figure 1, it looks like the confidence intervals around treatment effects are generally strongly overlapping across the entire x-axis of baseline risk. The one exception is the STAR dataset, where the lowest and highest point estimate have barely non-overlapping confidence intervals. As a result, another story consistent with the data is that effects are generally not so heterogeneous as conventional wisdom from the econometrics literature has it.


(2)

The results in Figure 1 are actually a fair bit more favorable towards risk-based targeting than the introduction of the paper had me believed. Treatment effects generally increase with risk. Targeting the 80th to 90th percentile of risk generally seems to capture high treatment effects across all datasets. So, another story could be that we should exclude the most extreme values of risk from targeting, but other than that risk-based targeting sort of works.


(3)

I found it rather confusing to have unnormalized utility values on the y-axis. After all the sample size is rather arbitrary and does not correspond to the population-level utility obtained if the policy maker were to implement the given approach. Along the same lines, I found it difficult to reconcile Figure 1 and Figure 2. See question below.


(4)

It would've been great to include datasets with real world confounding rather than the simulated confounding. My experience is that existing CATE estimation methods don't do very well in non-RCT settings. Might there be an advantage to risk-based targeting in non-RCT data?


Suggestion:

It seems to me that the story is much less certain than the abstract and introduction make it sound. I would therefore appreciate it if you could indicate a greater level of epistemic uncertainty in the writing throughout. I don't think this would hurt the paper at all. As is, though, I'd worry that your writing suggests the question is essentially closed conclusively which would actually discourage additional work in this direction.

**Questions:**

Visually, it seems difficult to reconcile the large difference between, say, Figure 2 top left panel and Figure 1 top left panel. It seems that on the TUP dataset, high risk is close to maximizing treatment effect, the highest values being around percentile 80. However, Figure 2 shows utility 15000 vs 5000 for treatment effect based vs risk vased. Can you explain?

Similarly, treatment effects for NSW seem to be largely constant around 1000, but Figure 2, second row left panel shows a massive advantage for treatment effect based targeting. This seems to be about a factor 7 (1.75 vs 0.25). Where do these large effect differences come from given that the treatment effect curve is essentially constant? None of the treatment effects seem to differ by more than a factor 2.

I tried to figure this out by looking through the code, but I couldn't find code that generated Figure 2. So, I don't actually know where it came from and what it shows. Maybe I missed it. On that note, it would be very helpful to clean up and document the code for the final version. As is, it's very hard to follow.

---

> ### Author Response · Authors · 2024-11-21
> **Reply to Reviewer 1**
>
> We thank you for your detailed review and constructive feedback. We address the weaknesses mentioned in the review in this comment and the questions in the subsequent comment. We have also edited the paper to incorporate the feedback we received and the edits are marked in red for easy distinction.
>
> Weaknesses:
> 1) Multiple reviewers pointed out that the interpretation of Figure 1 is ambiguous and suggest various alternative stories consistent with the plot. We agree with the reviewers and thank them for raising this point -- all of the interpretations that the reviewers mention do find some support in Fig 1 due to the wide confidence intervals. However, our main conclusions are driven by Fig 2, where we are able to draw much more statistically precise conclusions. We have edited the text to frame Fig 1 just as suggesting hypotheses about the effects of risk based targeting, which Fig 2 addresses with more precision/statistical significance. Here is the revised text discussing Fig 1:
>
> The estimated relationship between baseline risk and treatment effect is variable across domains. In most domains, the point estimate shows a general upward trend, indicating that individuals at greater risk benefit more (on average) from treatment. However, in the NSW domain, the point estimate is essentially flat. In addition, the confidence intervals are wide for all domains and there is very little statistically significant significant evidence in favor of high-risk individuals benefiting more. Wide confidence intervals reflect that there is significant variance in the pseudo-outcomes estimated for different individuals at the same level of baseline risk. That is, there is a great deal of variance in our estimated treatment effects that is not explained by baseline risk. From these results, we form two hypotheses. First, that risk-based targeting should, in most domains, perform better than a random allocation, since the point estimates generally show larger average effects at higher baseline risk. Second, that there is room to improve on risk-based targeting via strategies that leverage some of the substantial variance in treatment effects that is unexplained by baseline risk. The next section provides more statistically precise tests of these hypotheses by comparing the welfare associated with each targeting policy (a single number, which can be quantified more precisely than the entire curve shown in Figure 1).
>
> We have also edited the introduction and conclusion of the paper to reflect this framing.
>
> Regarding the specific concern you raise that "another story consistent with the data is that effects are generally not so heterogeneous as conventional wisdom from the econometrics literature has it": This alternative is ruled out by Fig 2, since we find (with statistical significance) that targeting based on heterogeneous treatment effects results in much higher welfare gains than random targeting, which would not be the case if treatment effects were largely homogeneous. Specifically, this is the comparison between the blue and green lines at the k = 0 point of the left-most column of Fig 2.
>
>
> 2) This connects to some of the points mentioned in Answer 1. The trend lines are more going up than down, so we do see some of the intuition for risk based targeting reflected in the data. To acknowledge this, we added the following sentence to the discussion: "We find that, in most domains, risk-based targeting results in higher welfare than a uniformly random allocation, confirming some of the intuition behind its widespread use by practitioners.” However, the confidence intervals are wide, reflecting that at any level of predicted risk, there is still a lot of variance in the individual-level outcomes, which a better targeting strategy may be able to leverage. This is confirmed in Fig 2, where we find statistically significant evidence that targeting based on estimated treatment effects is typically much more effective (comparison between the blue and orange lines at the k = 0 point of the left-most column of Fig 2).
>
>
> 3) Thanks for this suggestion. We have changed the y axis of Figure 2 to give the normalized utility. Regarding Figure 1 vs 2, we answer below.
>
>
> 4) Unfortunately none of the RCT datasets we use were accompanied by a matching observational cohort (a very unusual situation in practice). If confounding in a given domain is worse than our simulations, this could give an advantage to risk-based targeting. To provide a strong comparison, our simulation does reflect a very adversarial mode of confounding, where individuals select into treatment depending on the actual value of their potential outcome. This can be seen as a strengthening of a typical mechanism for confounding, where individuals are selected for treatment based on unobservable characteristics that are merely correlated with their potential outcomes. We edited the paper to explain this motivation.

---

> > ### Author Response · Authors · 2024-11-21
> > **Reply to Reviewer 1**
> >
> > In this comment, we respond to the questions posed:
> >
> > The key explanation is that the trend-lines Figure 1 only reflect one possible source of variation in treatment effects: variation based on baseline risk. Even when the curve is flat, as for NSW, there may be other sources of variation that targeting based on heterogeneous treatment effects can leverage. This is reflected in the wide confidence intervals in Figure 1, which arise because there is substantial variation in the estimated treatment effects for individuals with the same level of baseline risk. Similarly, for the TUP dataset, the discrepancy between Figure 1 and 2 just means that the majority of heterogeneous treatment effects are driven by characteristics orthogonal to baseline risk, even if a portion do correlate with baseline risk.
> >
> > Regarding the code: we will simplify/document it as suggested, for release along with the paper.
> >
> > We thank you again for your feedback.

---

> ### Author Response · Authors · 2024-11-25
>
> With the discussion period ending soon, we'd like to check if the above revisions to the paper address your concerns. Particularly since much of the discussion of this paper centers on the interpretation/framing of the results, we're hoping to make the most of the opportunity to improve the paper!

---

> > ### Comment · Reviewer_4SA7 · 2024-11-25
> >
> > They partially did, yes. Thanks for the helpful response! In particular, your point about the interpretation of Figure 1 is fair. There is no contradiction between the two figures.
> >
> > The source of my misunderstanding is also a remaining issue with Figure 1. The confidence intervals could be wide because there isn't enough data to make the claim, or because treatment effect is weakly correlated with predicted risk. I now understand that you're trying to argue the latter not the former.
> >
> > I'd say it's still worth clearing up the confusion. A more helpful alternative to Figure 1 might directly compare two curves: (a) treatment effect vs predicted risk, (b) treatment effect vs  predicted treatment effect. Curve (a) is essentially what you have in Figure 1, while curve (b) speaks to the predictability of treatment effects via your causal inference method. Curve (b) should look like a diagonal line with confidence intervals around it. Putting these two together should ideally make it visually obvious that targeting based on predicted treatment effects is superior to risk based targeting without even going through the utility calculation. Does this make sense?
> >
> > Please check some of the red text in your revision for spelling. There are some mistakes, e.g.:
> > "On some questions, are results are subject to greater uncertain"
> >
> > In any event, this is a good paper that is headed for acceptance, which I support!

---

### Meta-Review · Area_Chair_2ZvC · 2024-12-20

**Metareview:**

The paper compares two targeting strategies for allocating limited resources in social welfare applications: risk-based targeting and treatment-effect-based targeting. Using data from five real-world randomized controlled trials (RCTs), the authors find that targeting based on treatment effect estimates can outperform risk-based targeting, even when treatment effect estimates are biased or when policymakers prioritize assisting high-risk individuals. The study highlights the potential benefits of applying causal inference methods in policymaking, emphasizing that treatment-effect-based approaches yield greater social welfare gains across various scenarios.

Overall, the reviewers acknowledge that the paper addresses an important research question, employs a rigorous approach, and has significant potential impact. The main concerns revolve around the discussion of the results (e.g., Figure 1 and the new results added in rebuttal, as highlighted by XFvr). Considering the potential impact of the research and its generally strong execution, I am inclined to recommend accepting the paper. If accepted, please carefully address the reviewer comments when preparing the final version.

**Additional Comments On Reviewer Discussion:**

There were concerns around the discussion of the results (e.g., Figure 1 and the new results added in rebuttal, as highlighted by XFvr), and the author responses have helped clarify the issues.

---

### Decision · Program_Chairs · 2025-01-22

Accept (Poster)